# SCM: A Searched Convolutional Metaformer for SAR Ship Classification

**Hairui Zhu, Shanhong Guo \*, Weixing Sheng and Lei Xiao** [ID]

School of Electronic and Optical Engineering, Nanjing University of Science and Technology,
Nanjing 210094, China; hairui.zhu@njust.edu.cn (H.Z.); shengwx@njust.edu.cn (W.S.); leixiao@njust.edu.cn (L.X.)
\* Correspondence: guosh@njust.edu.cn

**Abstract:** Ship classification technology using synthetic aperture radar (SAR) has become a research hotspot. Many deep-learning-based methods have been proposed with handcrafted models or using transplanted computer vision networks. However, most of these methods are designed for graphics processing unit (GPU) platforms, leading to limited scope for application. This paper proposes a novel mini-size searched convolutional Metaformer (SCM) for classifying SAR ships. Firstly, a network architecture searching (NAS) algorithm with progressive data augmentation is proposed to find an efficient baseline convolutional network. Then, a transformer classifier is employed to improve the spatial awareness capability. Moreover, a ConvFormer cell is proposed by filling the searched normal convolutional cell into a Metaformer block. This novel cell architecture further improves the feature-extracting capability. Experimental results obtained show that the proposed SCM provides the best accuracy with only $0.46 \times 10^6$ weights, achieving a good trade-off between performance and model size.

**Keywords:** synthetic aperture radar; ship classification; deep learning; transformer; network architecture searching

## 1. Introduction

With the increasing number of ships on the oceans, efficient classification of ships has become vital [1,2]. Synthetic aperture radar (SAR) is a popular type of positive detection equipment used in modern ocean surveillance. SAR can work in all weathers and at all times [3]; thus, ship classification with SAR images has become a research hotspot in remote sensing.

Traditional SAR ship classification methods depend on feature extraction and classification using manual design, including support vector machine (SVM) [4], decision tree [5], random forest [6], Bayesian classifier [7], Adaboost [8], etc. However, these methods require large amounts of time and labor regarding their design, testing, analysis, and verification. The robustness and accuracy of these methods may significantly decrease in actual application scenarios with complex ship shapes, high sea conditions, and low image quality.

Recently, deep learning has achieved many excellent results in several fields of computer vision, including image processing [9,10] and pattern recognition [11,12]. Many studies have been conducted on applying deep learning to SAR ship classification. Compared to conventional methods, convolutional neural network (CNN)-based methods result in a remarkable boost in accuracy. Based on advanced training technologies, suitable natural optical image object classification models can operate SAR ship classification tasks [13–15]. For high accuracy, another more common method is to manually design networks for SAR data [16,17]. Furthermore, traditional features, such as histograms of oriented gradients (HOGs) and polarization coherence, can be integrated into neural networks to further improve performance [18,19].

Review of existing SAR ship classification networks suggests that most of the networks that are transplanted from computer vision or that are handcrafted have unsatisfactory

accuracies on SAR data or have an unfeasible model size for embedded devices. To further improve efficiency, a novel searched convolutional Metaformer (SCM) is proposed for SAR ship classification in this paper. The design of the SCM starts from a baseline network searched in a small SAR ship dataset. We improve the network architecture searching (NAS) algorithm. Considering the mobile and embedded platforms used in SAR tasks that require a high prediction efficiency, we improve the baseline network with transformer encoders and efficient convolutional cells. The SCM achieves a good trade-off between performance and computational complexity with a very small size.

Partial channel connection differentiable architecture searching (PC-DARTS) [20] treats the search as a bi-level optimization problem and updates the target network architecture by gradient descent [21]. The target network produced by the original PC-DARTS contains too many weight-free operations, which leads to low feature extraction capability. After an analysis of the involved SAR ship dataset, the small number of available samples, the poor quality of the SAR images, and the high similarity of the different ship categories result in the super net learning little about the features. Weight-free operations have a very high probability to be chosen at the early stage of searching. This problem is unsolvable automatically even at the end of searching, which leads to a target network with poor performance. Hence, progressive data augmentation partial channel connection differentiable architecture searching (PDA-PC-DARTS) is proposed to obtain a target network with strong feature excretion capability on a small size SAR ship dataset with poor image quality. This paper employs data augmentation and progressive learning to divide the whole search into three stages. A different data augmentation policy is assigned to a searching stage, so the target architecture is optimized progressively. The data augmentation can expand sample diversity. Progressive learning enhances the gradient learning of the super-net and architecture parameters. Our target network searching in the low-quality small-size SAR ship dataset shows improved performance.

Obviously, the performance of the target network has a great deal of room for further improvement. Networking scaling is the most common way to improve the performance of a searched network architecture. However, network scaling substantially increases the number of weights and computational operations. Considering the application scenario of SAR ship classification using mobile or embedded devices, we have to give up network scaling and choose an efficient way to improve performance. The convolutional operations and classifier in the target network are updated. The basic blocks of convolutional operations used in NAS are switched from depth-wise separable convolution (DSCONV) to the EfficientNet version inverted residual block (MBCONV) [22], which can improve the efficiency and accuracy. A transformer classifier is connected to the searched cells, which contributes to a searched convolutional transformer (SCT). The searched cells play the role of tokenizers instead of common simple tokenizers, such as a single fully connected layer [23] or stacked vanilla convolution layers [24]. The searched CNN has the advantage of processing two-dimensional data and the transformer classifier provides better learning capability. The SCT can combine the advantages of both parts and improve the performance of ship classification.

A transformer has a higher computational complexity than a CNN. Usually, transformer–CNN mixed networks in computer vision employ many transformer-based operations as the main computational module, which results in high computational complexity. In our approach, the performance of the CNN part on SAR ship data is guaranteed by NAS. Hence, the CNN is the main part in the proposed network. We only use two transformer encoders as an additional module and maintain the computational complexity of the whole network at a low level.

Then, following the concept of a Metaformer [25] and modern CNNs, a ConvFormer cell is proposed as the novel cell architecture. Studies have found that transformers have advantages for both self-attention and architecture. Hence, we convert a searched convolutional cell into a Metaformer block by referring to the transformer encoder architecture.

The token mixer in the ConvFormer cell is searched from the SAR ship dataset. Hence, the ConvFormer cell has a strong feature extraction ability on SAR ships.

SCM is a combination of SCT and ConvFormer cells. The experimental results on the OpenSARShip dataset show that SCM achieves 82.06% accuracy with $81.46 \times 10^6$ multiply-accumulate operations (MAdds). Compared with existing handcrafted learning-based methods and networks from computer vision, SCM outperforms them by 3% in accuracy and reaches a state-of-the-art level. To verify the generalization capability of the searched architecture, we train the proposed network on a more challenging classification task from the FUSARship dataset without prior research, resulting in a good performance of 63.90%. SCM only has $0.46 \times 10^6$ weights. In addition, the searching and training of SCM can be operated with only a single RTX3060.

We summarize the main contributions of the paper as follows:

1. A novel mixture network, SCM, is proposed for SAR ship classification. This network is built with the help of NAS and a transformer, which have the characteristics of high accuracy and small model size.
2. PDA-PC-DARTS is proposed to explore a high-performance target network in the poor quality, small sample number SAR ship dataset.
3. For searching, DSCONV is replaced with MBCONV to achieve better efficiency.
4. SCT is proposed by combining NAS, CNN, and a transformer, which improves the classification accuracy.
5. A novel cell architecture is proposed to further enhance the accuracy by filling a searched node into a Metaformer block.

The rest of this paper is organized as follows: Section 2 reviews related work. In Section 3, the SCM is described in detail. Section 4 describes the experiments. The results and ablation studies are described in Section 5. A discussion is provided in Section 6. Section 7 concludes the paper.

## 2. Related Work

### 2.1. CNN-Based SAR Ship Classification

In contrast to natural optical image object classification, samples of SAR ship classification show many special features, such as single channel, low quality, and simple background. CNN-based SAR ship classification methods have two main aspects: improving computer vision networks on SAR ship data and designing networks for SAR ship data. Table 1 summarizes some published CNN-based SAR ship classification methods.

Directly applying computer vision models to SAR ship data can give unsatisfactory results. Improving the adaptability of computer vision models on SAR data is the key to the first approach. A SAR ship classification method based on transfer learning was presented, where some layers of the visual geometry group network (VGGNet) pretrained on the ImageNet dataset, which is a popular public natural optical image object classification dataset, were retrained on SAR data [13]. VGGNet has a large mode size and high computational complexity because it was designed for optical object classification, which is a very difficult task in computer vision. A hybrid channel feature loss was designed to improve the training of ship SAR images [14]. This dual-polarization classification method also employed VGGNet and achieved good accuracy. The model size and computational complexity of this method are much larger than many methods using networks designed for SAR ship data. Thanks to the development of deep learning, a dual-polarization classification method based on mini-hourglass region extraction and a dual-channel efficient fusion network has achieved good accuracy with moderate computational complexity [15]. Many advanced technologies based on deep learning have been integrated into this method, which has boosted its efficiency. However, the core of this method is EfficientNet, whose target platform is a desktop. Compared with some networks designed for SAR aiming at mobile or embedded platforms, this method has relatively higher computational complexity.

Networks designed for SAR ship data are different to networks designed for computer vision. A sequential CNN for high-resolution SAR ship images has been designed, including ten convolutional layers and three fully connected layers [16]. Experimental results on the dataset obtained from the Gaofen-3 satellite showed that this CNN had good accuracy over many ship categories. However, this method has very higher computational complexity compared to VGGNet-based methods. Furthermore, the number of weights in this method is not suitable for embedded devices. Traditionally handcrafted feature extraction can be integrated into neural networks. A combination of HOGs, principal component analysis (PCA), spatial attention, and a CNN was proposed and referred to as the network with HOG feature fusion (HOG-ShipCLSNet) [18]. This method contains two data flows. For one data flow, the features in HOGs are extracted by PCA. For the other data flow, convolutional layers with spatial attention are used to extract features from the original SAR image at different scales. Concatenated feature vectors from the two flows are fed to a linear classifier. Compared with other CNN-based SAR ship classification methods, HOG-ShipCLSNet has low computational complexity and high performance. Two disadvantages of this method are the large model size and the need for many additional computations outside of the neural network. A dual-polarization SAR ship classification network was constructed with a channel-wise attention fusion module, a squeeze-and-excitation (SE) module, and a Laplacian feature pyramid [19]. This squeeze-and-excitation Laplacian pyramid network was dubbed "SE-LPN-DPFF". The dual-polarization SAR images and the polarization coherence are the inputs. This method shows good performance and depends largely on dual-polarization with a large prediction latency. Many SAR products do not provide dual-polarization. Single-polarization-based methods can work on dual polarization data with simple fusion. Hence, compared to single-polarization-based methods, SE-LPN-DPFF has limited scope for application.

**Table 1.** Summary of CNN-based SAR ship classification methods.

| Method | Use Traditional Feature | Network Architecture Designed for SAR Ships | Model Size | Computational Complexity | Dual-Polarization Only |
|---|---|---|---|---|---|
| Finetuned VGGNet [13] | × | × | Large | High | × |
| VGGNet With Hybrid Channel Feature Loss [14] | × | × | Large | High | √ |
| Mini Hourglass Region Extraction and Dual-Channel Efficient Fusion Network [15] | × | × | Moderate | Moderate | √ |
| Plain CNN [16] | × | √ | Large | High | × |
| HOG-ShipCLSNet [18] | √ | √ | Large | Low | × |
| SE-LPN-DPFF [19] | √ | √ | | | √ |

### 2.2. Network Architecture Searching

NAS is a kind of algorithm that automatically designs network architectures for predefined purposes. A manually designed network architecture always requires a long development period with low labor efficiency. Furthermore, unnecessary interventions from involved researchers may damage these handcrafted networks.

Hence, to improve upon manually designed networks, NAS has become one of the most important research areas in computer science. NAS algorithms can be divided into three main categories reflecting their different approaches to exploring the searching

space: reinforcement-learning-based searching, evolutionary-based searching, and gradient-based searching.

Reinforcement-learning-based searching is the most important area to date and has had a remarkable impact on all NAS algorithms. An outside meta-controller is defined to generate the network architectures. The controller is taught to act following predefined goals through the use of reinforcement learning. Application of the concept of a node cell prevents unfinishable search with an exaggerated large searching space for the whole network [26]. The target network can be built with several stacked searched cells sharing the same prediction graph, which means the output of searching changes from a whole network to a few cells, significantly reducing the computational complexity. Weight-sharing searching significantly reduces the time consumption for searching [27]. Clearing trained weights is forbidden, which eliminates the time used to retrain from scratch at the evaluation stage.

Evolutionary-based searching focuses on applying evolutionary algorithms and genetic operations on network architectures with predefined targets. With the help of encoding technology, neural networks can be presented as architecture gene queries [28]. Furthermore, some network architectures for special uses, such as resisting noise [29], can be found with this approach.

Gradient-based searching was established based on converting architecture searching into an optimization problem and achieves excellent searching efficiency. Differentiable architecture searching (DARTS) was the first gradient-based searching algorithm [21]. Searching is defined as a bi-level optimization problem which finds the target network by applying gradient descent. In the meantime, both node cell searching and weights sharing are utilized. All candidate operations are packaged into a super-net and can be trained together, which significantly decreases the search time. However, DARTS still has server-level hardware requirements and can be criticized for filling many weight-free operations in target networks. PC-DARTS is a personal-computer-level NAS [20]. Randomly sampled channels can effectively reduce the computational complexity and memory occupation, while the hit rates of weight-free operations with constant output are decreased to a moderate state.

### 2.3. Efficient Convolutional Blocks

Due to popular modern CNNs being built with convolutional blocks, many researchers are working on modifying convolutional blocks to improve the efficiency of CNNs. Related studies are being conducted on pure convolutional blocks and additional module-equipped blocks.

An efficient pure convolutional block usually contains one or several convolutional layers, activations, normalizations, and shortcut connections. DSCONV is a well-known efficient convolutional block, where a normal convolution is replaced by a depth-wise convolution and a point-wise convolution [30]. DSCONV has an advantage in having very low computational complexity and has become a pioneer for the implementation of CNNs in mobile devices. In depth-wise convolution, each kernel is only connected to an input channel. Point-wise convolution can be considered as a normal convolution whose kernel size is set as 1. However, when the number of input channels is small, the performance of DSCONV is significantly reduced. MobileNetv2 version MBCONV has an inverted residual architecture to alleviate this problem [31]. This architecture first expands the dimension of the feature maps with a point-wise convolution. After computing the depth-wise convolution, the dimension is projected back with another point-wise convolution.

The second approach towards creating efficient convolutional blocks is by inserting additional modules into blocks to improve the performance with a small cost in computational complexity. Inserting the SE module into the residual blocks can significantly improve accuracy [32]. An SE module contains two fully connected layers and one global average pooling layer. After processing of an SE module, the key channels are enhanced, which can be considered as a kind of channel-level attention. Based on MobileNetv2, an improved version of MBCONV, that contains the SE module and uses Swish for activation, is the basic block of EfficientNet [22]. With the help of NAS, EfficientNet achieves a good trade-off between computational complexity and performance. However, the model size and computational complexity of the smallest member of the EfficientNet family is unaffordable for most embedded devices because EfficientNet is designed for middle- and large-sized devices.

*2.4. Networks Related to Transformers*

After the great success of transformer-based models in natural optical image classification and detection, researchers are increasingly working with transformers on two-dimensional data. The three main types of networks related to transformers can be summarized as transformer-based networks, CNN–transformer mixed networks, and Metaformer-based networks.

Transformer-based networks use self-attention to process features. Vision transformer (ViT) is one of the pioneers in this approach [23]. An image is divided into 286 ($14 \times 14$) patches of $16 \times 16$ pixels. After tokenization and positional embedding, tokens are fed into stacked transformer encoders. The Hierarchical Vision transformer using Shifted Windows (Swin) uses cross-attention to achieve CNN-like multi-level feature extraction and provides multi-scale feature maps [33]. Self-attention involves processing the features within its receptive field and sensing the whole feature map with slicing windows, which decreases the computational complexity.

Some studies on integrating CNNs and transformers have revealed that mixture networks exhibit advantages from both sides. CoAtNet-equipped MBCONV blocks and transformer encoders exhibit more powerful performance on some small-size datasets, indicating that CNNs and transformers exhibit differences in learning capability and generalization capability [34].

The compact convolutional transformer (CCT) is aimed at lightweight model applications [24]. A classic CNN with several normal convolutional layers and max pooling layers is used as a tokenizer, which can efficiently extract features and reduce some of the requirements for transformer parts regarding feature extraction. The CCT outperforms most transformer-based networks and other mixed networks in model size and computational complexity.

Metaformer-based networks focus on the characteristics of the transformer encoder architecture and use other operations to act as token-mixers instead of self-attention. Poolformer uses average pooling, which has no weights, as the token-mixer [25]. Compared to transformer-based networks with multi-head self-attention, Poolformer has much lower computational complexity. Another Metaformer uses multilayer perceptron (MLP) as the token-mixer and achieves acceptable results on image classification tasks [35]. Although a gap in performance between this method and the most advanced CNNs can be observed, its special architecture can significantly help future investigations into interactions in a neural network. Figure 1 displays the general concept of the Metaformer block and the architecture of the transformer encoder and poolformer block.

Norm: Normalization
FC：Fully Connected (Linear)
LN: Layer Normalization
MHSA:Multi-head Self-attention
GN: Group Normalization

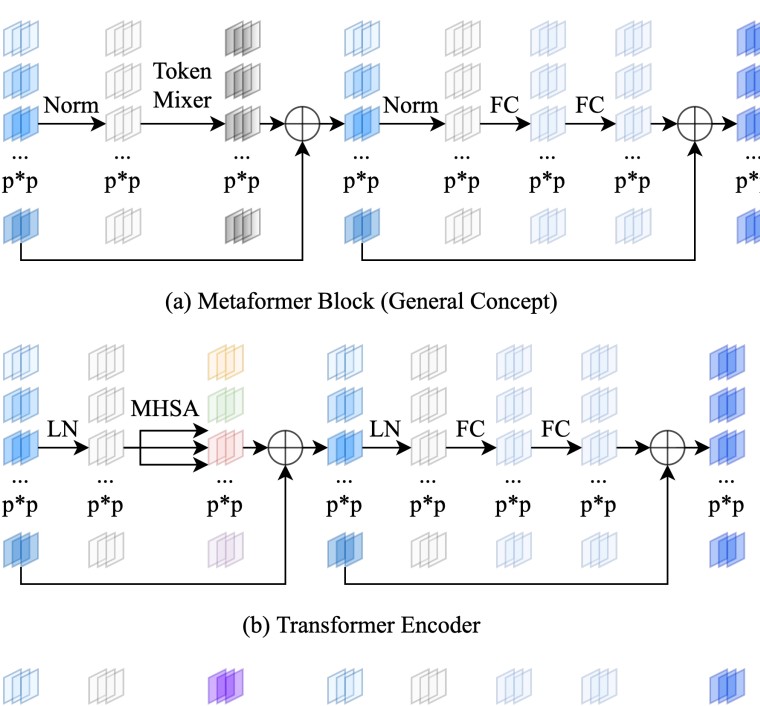

**Figure 1.** Architecture of a Metaformer block and its specific applications.

### 2.5. SAR Image

Radars transmit modulated radio signals and receive the reflected signals from objects to detect targets, which is very different from cameras, which use natural light or infrared radiation [36]. The wavelength of the radio frequency signal and the polarization processes of radar can affect the SAR images produced. In addition, SAR technology itself and the ocean also present problems for the SAR imaging of ships.

Radar detection waves are able to penetrate, which allows SAR to operate in all weathers. Different wavelengths have different penetration distances, which could affect SAR images. The X-band has a wavelength of three centimeters (cm), the C-band has a wavelength of 6 cm, the L-band has a wavelength of 24 cm, and the P-band has a wavelength of 65 cm. Usually, a signal in the C-band can only penetrate the upper levels of a forest's canopy and can show much roughness scattering combined with some volume scattering. A signal in the P-band or L-band could result in more volume scattering and double-bounce scattering [37]. The C-band has been widely applied in many SAR-equipped satellites, such as Sentinel-1 and Gaofen-3.

Different polarization processes show different features in the resultant SAR images. When describing radar polarization processes, the letters H and V stand for horizontal and vertical polarization, respectively. The transmitter and receiver of a radar may employ polarization processes; hence, the complete polarization process of an SAR image can be described in two letters. VV polarization means both the transmitter and receiver use vertical polarization. This polarization can be more easily influenced by sea clutter. If the transmitter and receiver are set with different polarization processes, VH or HV will be used. These polarizations are susceptible to volume scattering. HH polarization has a stronger double-bounce [37].

SAR images of the sea do not show distance distortion, which is common in optical images, but other problems remain. Geometric distortion can be generated in SAR images by deviation of the detecting geometry, refraction, turbulence, etc. Many SAR detectors are present in high-speed aircraft or where the target is moving rapidly [38]. The relative speed leads to range migration [39]. In addition, speckle noise is generated by the SAR imaging technology itself and is unavoidable in SAR images [40].

## 3. Searched Convolutional Metaformer

Convolutional cells and a transformer classifier are integrated together in the proposed network for SAR ship classification. The overall architecture of the SCM is shown in Figure 2 and Table 2. The SCM contains four searched convolutional cells, two proposed ConvFormer cells, and one transformer classifier. To achieve a small model size and low computational complexity, a very small number of initial channels is selected and the sizes of the feature maps are reduced many times in the SCM.

In this section, convolutional cells are introduced, with details covered including cell architectures, updated basic blocks, and the proposed searching algorithm which is designed for SAR data. Then, the transformer classifier and proposed ConvFormer cell are described.

**Table 2.** Details of the SCM network architecture.

| Name | Type | Input(s) | Input Shape(s) | Output Shape |
|------|------|----------|----------------|--------------|
| Stem | Normal Convolution | SAR Image | $1 \times 100 \times 100$ | $12 \times 48 \times 48$ |
| Cell1 | Normal Cell | #1 Stem<br>#2 Stem | #1 $12 \times 48 \times 48$<br>#2 $12 \times 48 \times 48$ | $16 \times 48 \times 48$ |
| Cell2 | Reduction Cell | #1 Stem<br>#2 Cell1 | #1 $12 \times 48 \times 48$<br>#2 $16 \times 48 \times 48$ | $32 \times 24 \times 24$ |
| Cell3 | Reduction Cell | #1 Cell1<br>#2 Cell2 | #1 $16 \times 48 \times 48$<br>#2 $32 \times 24 \times 24$ | $64 \times 12 \times 12$ |
| Cell4 | Normal Cell | #1 Cell2<br>#2 Cell3 | #1 $32 \times 24 \times 24$<br>#2 $64 \times 12 \times 12$ | $64 \times 12 \times 12$ |
| Cell5 | ConvFormer Cell | #1 Cell3<br>#2 Cell4 | #1 $64 \times 12 \times 12$<br>#2 $64 \times 12 \times 12$ | $64 \times 12 \times 12$ |
| Cell6 | ConvFormer Cell | #1 Cell4<br>#2 Cell5 | #1 $64 \times 12 \times 12$<br>#2 $64 \times 12 \times 12$ | $64 \times 12 \times 12$ |
| Transformer Classifier | Compact Transformer Classifier | Cell6 | $64 \times 12 \times 12$ | 3 |

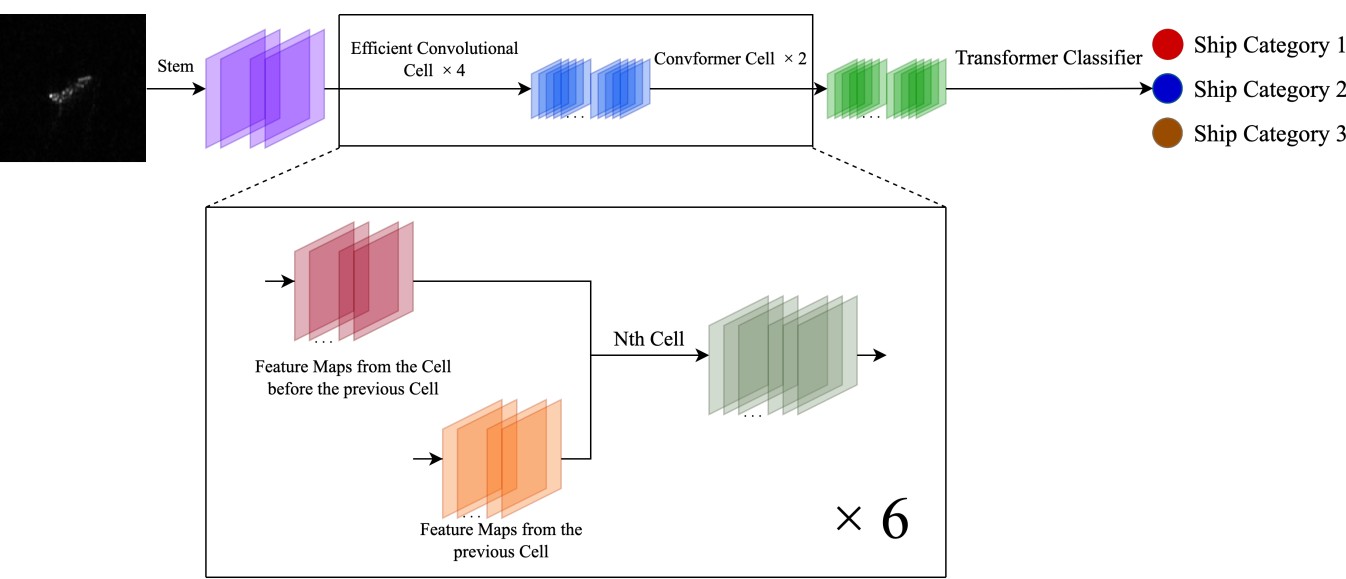

**Figure 2.** Overall network architecture of the SCM.

### 3.1. Efficient Convolutional Cells

The efficient convolutional cells in this paper include normal cells and reduction cells, which are shown in Figure 3. A normal cell is constructed with five efficient $5 \times 5$ convolutional operations, two efficient $5 \times 5$ dilated convolutional operations, and one efficient $3 \times 3$ convolutional operation. A reduction cell contains four efficient $5 \times 5$ convolutional operations, two efficient reduction $5 \times 5$ convolutional operations, one efficient $3 \times 3$ convolutional operation, and one efficient reduction $3 \times 3$ convolutional operation. All the efficient operations use MBCONV as the basic block instead of DSCONV from DARTS and PC-DARTS. The type and the connections of each operation in our efficient convolutional cells are determined by the proposed PDA-PC-DARTS.

Progressive Network Architecture Searching

Inspired by studies about progressive learning and data augmentation [41], PDA-PC-DARTS is proposed based on PC-DARTS. The search is divided into three stages with different data augmentation policies, which makes the super-net learn better with low-quality SAR ship data. The convolutional cells in the SCM are obtained with the help of PDA-PC-DARTS.

Original PC-DARTS divides the training set into two parts equally. One part is used to train the super-net and the other is used to optimize the architecture parameters. Meanwhile, only a moderate data augmentation policy is employed during the entire search. As a result, weight-free operations, such as pooling, are highly likely to be selected in SAR ship data. The target network is unable to produce a satisfactory performance.

In this paper, more complex data augmentation policies are used. In addition, both super-net training and architecture parameter searching are conducted on the whole training set, which can improve the data utilization rate and the search performance. Following the concept of progressive learning, PDA-PC-DARTS lets the super-net learn small-size images with weak transformations at the beginning stage. Then, large images with a strong data augmentation policy are used in training. Considering the characteristics of SAR images, the search progress contains three stages where data augmentation technologies are used to change the difficulty of learning and increase the variety of data. Hence, the super-net in PDA-PC-DARTS can learn more about samples and produces a target network with better performance. The data augmentation policy in each stage is shown below:

In the first stage, the size of the input images is $76 \times 76$, obtained by center cropping and Bicubic. The first stage consists of 12 epochs.

In the second stage, the size of the input images is $84 \times 84$, obtained by random cropping and use of Bicubic. Additional sample transformation methods include random horizontal flipping and cutout [42]. The second stage consists of 12 epochs.

In the third stage, the size of the input images is $100 \times 100$, obtained by random cropping. Additional sample transformation methods include random horizontal flipping, vertical flipping, rotation, and cutout. The third stage consists of 36 epochs.

Figure 4 demonstrates the different outputs with all three data augmentation policies used in the different searching stages. Notice that human eyes are not sensitive to dark pixels in the original SAR images. For a better viewing experience, a row of enlarged images with increased lightness is shown in Figure 4.

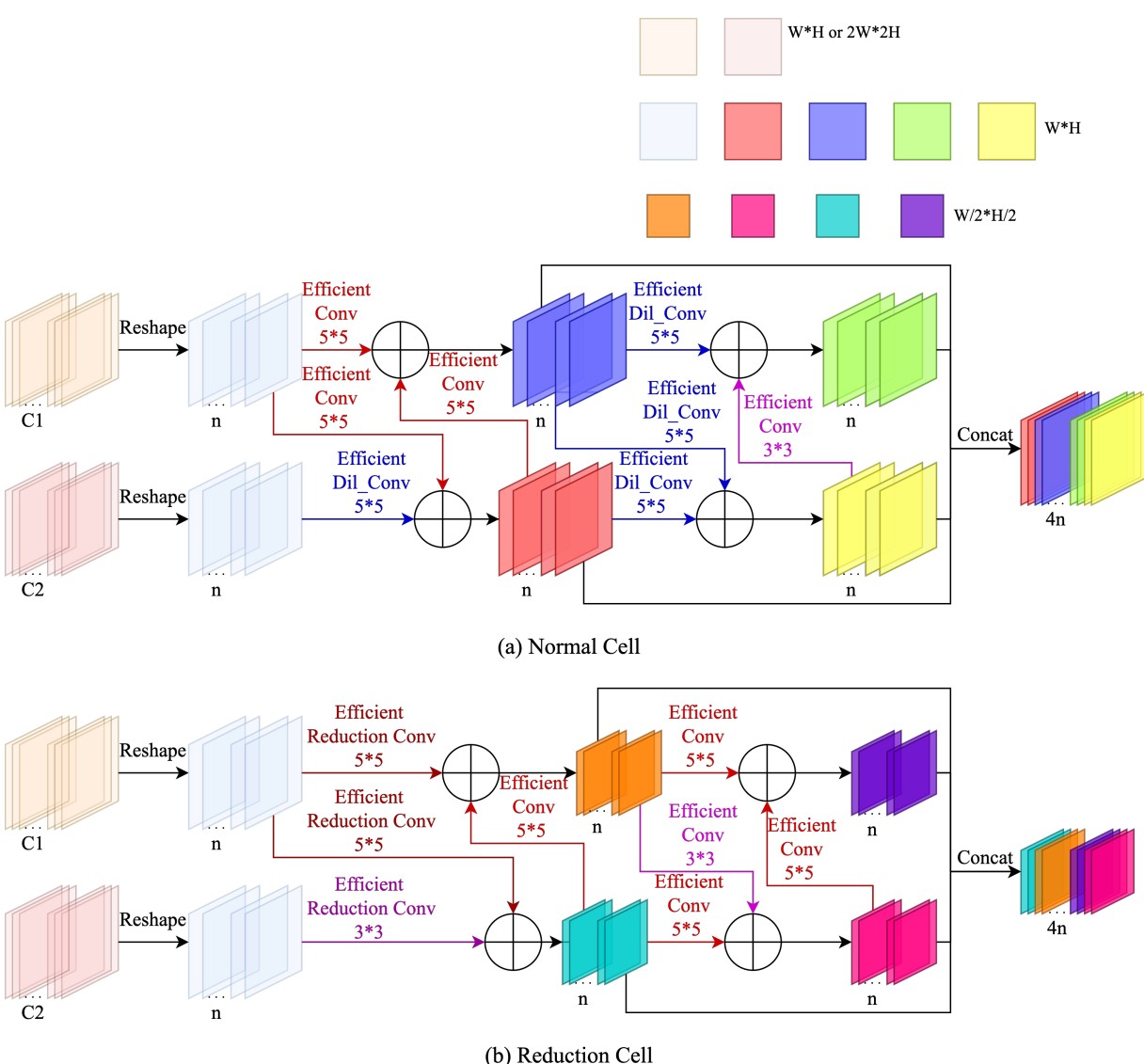

**Figure 3.** Architectures of searched cells. Efficient convolutional and dilated convolutional operations do not change the shape of feature maps. Efficient reduction convolutional operations reduce the height and width of feature maps by half.

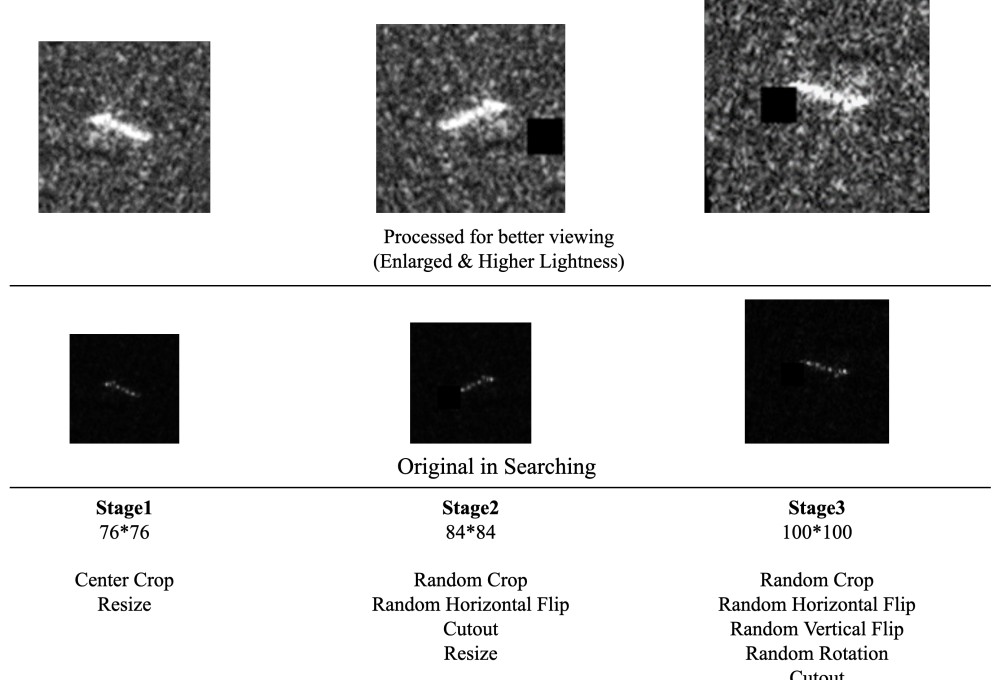

**Figure 4.** Demonstration of different data augmentation policies.

The mixture operation used in PDA-PC-DARTS is the same as the original random partial connection mixture operation in PC-DARTS, which ensures very low computational complexity and high memory efficiency. Taking the information propagating from node *i* to node *j* as an example, the mixture operation $f_{i,j}(.)$ is shown below:

$$f_{i,j}(x_i; S_{i,j}) = \sum_{o \in O} \frac{\exp\{\alpha_{i,j}^o\}}{\sum_{o' \in O} \exp\{\alpha_{i,j}^{o'}\}} o(S_{i,j} \times x_i) + (1 - S_{i,j}) \times x_i \tag{1}$$

where $O$ is the predefined searching space, $o(.)$ is the candidate operation, $x_i$ is the output of node *i*, $\alpha_{i,j}^o$ is the weight for choosing the candidate operation $o(.)$, and $S_{i,j}$ are randomly sampled channels.

Random sampling can directly reduce the consumption of computational resources and memory, but the search progress is highly likely to be unstable. Similarly, we apply the solution from PC-DARTS. Edge normalization is used to stabilize searching. The output of node *j* in the search with edge normalization can be described as:

$$x_j = \sum_{i<j} \frac{\exp\{\beta_{i,j}\}}{\sum_{i'<j} \exp\{\beta_{i',j}\}} f_{i,j(x_i;S_{i,j})} \tag{2}$$

where $\beta_{i,j}$ is the edge normalization coefficient.

### 3.2. Efficient Convolutional Operations

DSCONV has the advantage of low computational complexity, but low accuracy performance can be observed when a small channel number is set. Recently, research on efficient networks has shown the EfficientNet version MBCONV has strong feature extraction and generalization capability [22,24]. To achieve better performance, we replace DSCONV with the EfficientNet version MBCONV as the basic block of candidate operations. The architecture of the EfficientNet version MBCONV block is represented in Figure 5.

PW CONV:Point-wise Convolution
DW CONV: Depth-wise Convolution
GAP: Global Average Pooling
FC: Fully Connected (Linear)

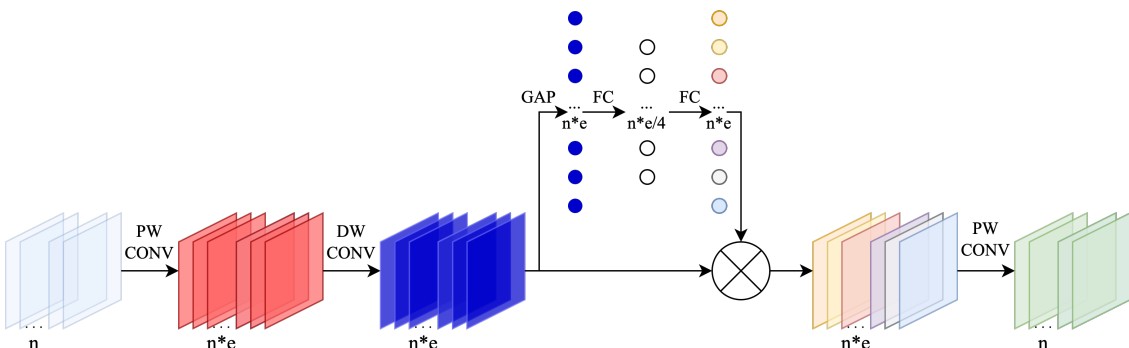

**Figure 5.** Architecture of EfficientNet version MBCONV block.

Each operation in the SCM consists of one or two stacked MBCONV blocks. To ensure the search space meets the requirements, our replacement keeps the features of the original candidate operations. The hyperparameters of the first depth-wise convolution in each of our efficient operations, such as the kernel size and dilation, are kept the same as the original operations. The details of the updated operation are shown in Table 3.

**Table 3.** Configuration of efficient convolutional operations.

| Original Operation | Efficient Operation | Original Block(s) | Updated Block(s) | Hyperparameters of Depth-Wise Convolution(s) |
|---|---|---|---|---|
| Conv $3 \times 3$ | Efficient Conv $3 \times 3$ | $2\times$ DSCONV | $2\times$MBCONV | #1 Kernel $3 \times 3$, Stride 1, Dilation 1<br>#2 Kernel $3 \times 3$, Stride 1, Dilation 1 |
| Conv $5 \times 5$ | Efficient Conv $5 \times 5$ | $2\times$ DSCONV | $2\times$MBCONV | #1 Kernel $5 \times 5$, Stride 1, Dilation 1<br>#2 Kernel $5 \times 5$, Stride 1, Dilation 1 |
| Dil_Conv $3 \times 3$ | Efficient Dil_Conv $3 \times 3$ | DSCONV | MBCONV | Kernel $3 \times 3$, Stride 1, Dilation 2 |
| Dil_Conv $5 \times 5$ | Efficient Dil_Conv $5 \times 5$ | DSCONV | MBCONV | Kernel $5 \times 5$, Stride 1, Dilation 2 |
| Reduction Conv $3 \times 3$ | Efficient Reduction Conv $3 \times 3$ | $2\times$ DSCONV | $2\times$MBCONV | #1 Kernel $3 \times 3$, Stride 2, Dilation 1<br>#2 Kernel $3 \times 3$, Stride 1, Dilation 1 |
| Reduction Conv $5 \times 5$ | Efficient Reduction Conv $5 \times 5$ | $2\times$ DSCONV | $2\times$MBCONV | #1 Kernel $5 \times 5$, Stride 2, Dilation 1<br>#2 Kernel $5 \times 5$, Stride 1, Dilation 1 |
| Reduction Dil_Conv $3 \times 3$ | Efficient Reduction Dil_Conv $3 \times 3$ | DSCONV | MBCONV | Kernel $3 \times 3$, Stride 2, Dilation 2 |
| Reduction Dil_Conv $5 \times 5$ | Efficient Reduction Dil_Conv $5 \times 5$ | DSCONV | MBCONV | Kernel $5 \times 5$, Stride 2, Dilation 2 |

Similar to DSCONV, point-wise convolution and depth-wise convolution play important roles in MBCONV. Point-wise convolution is a normal convolution with a kernel size of 1. Furthermore, depth-wise convolution is a modified convolution where each channel is connected to only one input channel. The equations of normal convolution and depth-wise convolution are shown below:

$$CONV_{C_{out}}(In) = B_{C_{out}} + \sum_{k=0}^{C_{in}-1} W_{C_{out}}^k \bigotimes In_k \tag{3}$$

$$DWCONV_{C_{in}}(In) = B_{C_{in}} + W_{C_{in}} \bigotimes In_{C_{in}} \tag{4}$$

where $W$ is the weight, $B$ is the bias, $In$ is the input, and $C$ indicates the number of channels.

A novel activation, Swish, is equipped in the EfficientNet version MBCONV. This function is obtained with the help of exhaustive and reinforcement-learning-based automatic search techniques and is characterized by smoothing, the first derivative, and non-monotonicity [43]. The computation of Swish is as follows:

$$Swish = In \times Sigmoid(In) \tag{5}$$

### 3.3. Transformer Classifier

Some studies have shown that combining a transformer and a CNN together is beneficial to tasks with two-dimensional data. To enhance the learning capability, the compact transformer classifier from CCT [24] is used as the classifier in the SCM.

The transformer classifier of the SCM only contains two transformer encoders. Firstly, convolution shows more advantages than transformer blocks in a small-size two-dimensional dataset. Secondly, the transformer encoder suffers more from overfitting and usually has higher computational complexity [34]. Lastly, the searched convolutional cells already have strong feature extraction capability in SAR ship images, and adding a small number of self-attention-based modules can help obtain improved learning capability. We believe that the ratio of convolutional operations to transformer blocks in the SCM leads to a better trade-off between accuracy and computational complexity. The classifier used in the SCM is shown in Figure 6, including one trainable positional embedding, two staked transformer encoders, one sequence pooling, and one fully connected layer.

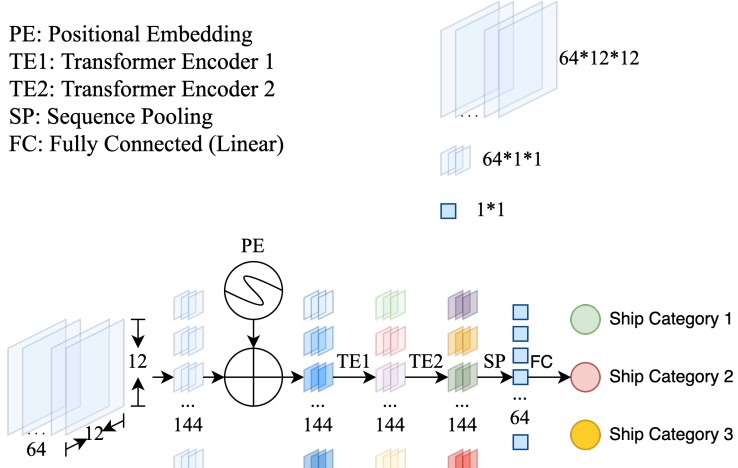

**Figure 6.** Architecture of the transformer classifier.

Our transformer classifier assigns a positional vector to each pixel of the input feature maps and processes information at the spatial level with transformer encoders. One multi-head self-attention mechanism, two normalizations, two residual connections, and two fully connected layers can be found in a transformer encoder. The multi-head self-attention mechanism can be thought of as the parallel computing of several self-attentions with a concatenated and transformed output, which can be calculated as:

$$MultiHead(Q, K, V) = [head_1, head_2, \ldots, head_h]W_0 \tag{6}$$

$$head_l = Softmax(\frac{Q_l K_l{}^T}{\sqrt{d_k}})V_l \tag{7}$$

$$Q_l = InW_l^Q \tag{8}$$

$$K_l = InW_l^K \tag{9}$$

$$V_l = In W_l^V \tag{10}$$

### 3.4. ConvFormer

Inspired by recent studies on network architecture [25], a novel cell, the ConvFormer cell, is proposed in this paper. The searched convolutional cells show higher power feature extraction capability than some simple layers or weight-free operations, such as MLP and pooling. Following the concept of a Metaformer, enhancement of the node cell architecture can further improve the performance of the SCM.

The ConvFormer Cell is designed by filling a Metaformer block. The searched efficient convolutional cell is used as the token mixer, which can combine the advantages of both NAS and Metaformer. The architecture of our proposed ConvFormer cell is shown in Figure 7. This cell adds two residual connections, two group normalization layers [44], and two fully connected layers activated by GeLu [45]. Notice that the residual connection requires that both inputs share the same shape. Hence, the token mixer in a ConvFormer cell is only a normal cell.

FC：Fully Connected (Linear)
GN: Group Normalization

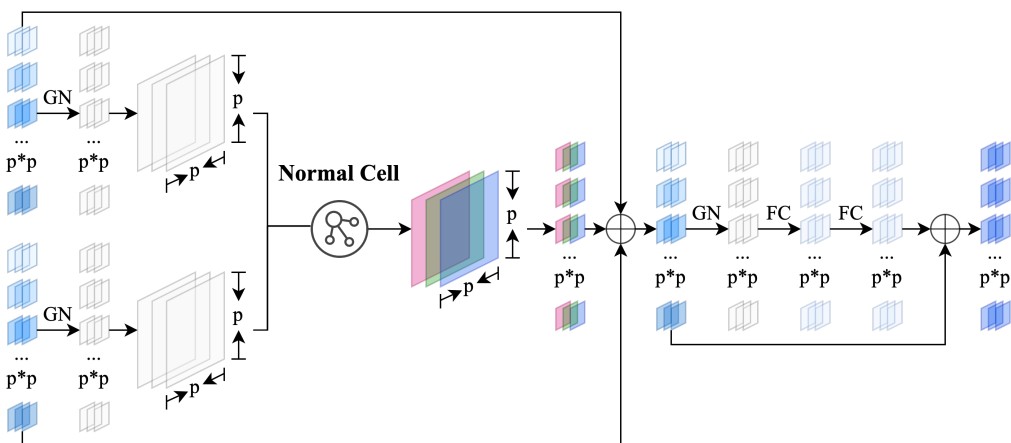

**Figure 7.** Architecture of the ConvFormer cell.

Metaformer networks with simple token mixers usually require a large number of Metaformer blocks to ensure satisfactory performance, which leads to a large model size. The performance of the SCM is guaranteed by searched convolutional cells. A very small number of ConvFormer cells is used in the SCM to obtain a small model. Furthermore, the SCM does not require an additional patch embedding module, the function of which can be replaced by convolutional operations. As a result, considering the trade-off between computational complexity and performance, we build an SCM with six cells by appending two ConvFormer cells to four stacked efficient convolutional cells. The channel number and type of each efficient convolutional cell are set according to the rules of DARTS.

## 4. Experiments

A series of experiments were conducted on the OpenSARShip dataset and FUSARShip to verify the superiority of the proposed SCM. A personal computer with an Intel core i5-11600 CPU, 16 G memory, and only a single RTX 3060 was used. The main software environment was an Ubuntu20.04 LTS operating system and the Pytorch framework [46] with the compute unified device architecture (CUDA) toolkit.

*4.1. Datasets*

4.1.1. OpenSARShip

OpenSARShip is an open-access dataset that has been widely used in many SAR studies [47]. A total of 11,346 SAR ship images from the ground range detected mode or the single look complex mode are contained in OpenSARShip. The size of a category in this dataset is distributed over a wide range. The single look complex mode provides both VV and VH data for a target with good range resolution. Bulk carriers, container ships, and tankers are selected to construct the classification task. The involved categories include approximately 80% of international routes [48].

To avoid the long tail problem, we follow the rule of HOG-ShipCLSNet [18] to take 70% of the least sample number in all three categories as the uniform number of the training targets for each category. Furthermore, the remainder of the targets are put into the test set. VV and VH data of the same target are treated as two samples and put in the same set. The sample number of each category is shown in Table 4; some samples are shown in Figure 8.

**Table 4.** Sample numbers of each category in the three-category classification task from OpenSARShip.

| Ship Category | VH Samples | VV Samples | Training Samples | Test Samples |
|---|---|---|---|---|
| Bulk Carrier | 333 | 333 | 338 | 328 |
| Container Ship | 573 | 573 | 338 | 808 |
| Tanker | 242 | 242 | 338 | 146 |



**Figure 8.** SAR ship samples in the three-category classification task from OpenSARShip. The targets in a column are the same.

4.1.2. FUSARShip

FUSARShip is another open-access dataset for SAR research and is built with the Gaofen-3 civil C-band fully polarimetric spaceborne SAR and related ship automatic identification systems [16]. Ships, non-ship targets, land, sea clutters, and false alarms are contained in this dataset. Although the number of ship categories in FUSARShip is large, the data sizes of the categories are much more unbalanced. A classification task is built with the bulk carriers, fishing ships, cargoes, and tankers, including both onshore and offshore scenarios.

Similarly, 70% of the least sample number in all four categories is set as the uniform number of training targets for each category. Moreover, the sample numbers of the cargo and fishing categories are much larger than the other categories. The testing sample number of each category is set to be the same as the third sample number in the involved categories. The sample number of each category is shown in Table 5; some samples are shown in Figure 9.

**Table 5.** Sample numbers of each category in the four-category classification task from FUSARShip.

| Ship Category | Total Samples | Training Samples | Test Samples |
|---|---|---|---|
| Bulk Carrier | 274 | 174 | 100 |
| Fishing | 787 | 174 | 100 |
| Cargo | 1735 | 174 | 100 |
| Tanker | 248 | 174 | 74 |

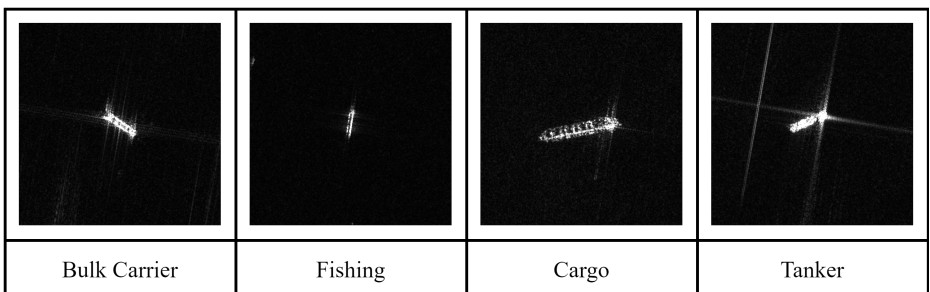

| Bulk Carrier | Fishing | Cargo | Tanker |

**Figure 9.** SAR ship samples in the four-category classification task from FUSARShip.

### 4.2. Data Preprocessing

OpenSARShip provides processed Uint8 images with different image shapes. To reduce the distortion effect of resizing, samples over $128 \times 128$ are center-cropped into $112 \times 112$. Meanwhile, samples smaller than $112 \times 112$ are resized to $112 \times 112$ by Bicubic. Different sample shapes are made uniform to $100 \times 100$ with cropping. In detail, random and center cropping are used in training and testing, respectively.

The samples from FUSARShip share the same image shape of $512 \times 512$. Hence, samples are first resized to $128 \times 128$. Then, cropping is used to create images with a size of $100 \times 100$.

### 4.3. Architecture Searching

The convolutional cells in the SCM are searched in the three-category classification task from OpenSARShip with the proposed PDA-PC-DARTS. The search is operated with a batch size of 32. The SGD optimizer [49] is used to update the super-net and architecture parameters. The learning rate is set with an initial value of 0.04 and a final value of 0.001. The cosine annealing scheduler changes the learning rate during training following the equation shown below:

$$\eta_t = \eta_{min} + \frac{1}{2}(\eta_{max} - \eta_{min})(1 + cos(\frac{T_{cur}}{T_{max}})) \tag{11}$$

where $\eta_t$ is the current learning rate, $\eta_{max}$ is the max learning rate, $\eta_{min}$ is the final learning rate, and $T_{cur}$ and $T_{max}$ indicate the current epoch and final epoch, respectively.

### 4.4. Network Training

To improve learning with a small number of available samples, the SCM is trained with data augmentation and multi-task learning technologies.

Data augmentation can significantly increase the variety of samples and has been proven to improve the learning of SAR data [18]. The applied transformations in the SCM training policy include random cropping, flipping, and cutout.

Usually, transformer encoders require a large amount of training data to obtain satisfactory learning, which has led to it being referred to as "data hungry". To reduce this phenomenon in training with the small SAR dataset, this paper takes an approach based on multitask learning to improve performance [50]. The loss in training the SCM is the sum of the distance loss and cross-entropy. To calculate the distance loss, a pair of patches from the last transformer encoder are randomly picked up. An MLP is used to output the distance between the patches. The L1 loss of the outputted value and the real value is the result of the distance loss. The equation for the distance loss is shown below:

$$L_{dr} = \sum_{s \in B} E_{(p_{i1,j1}, p_{i2,j2})\ G_s} [|(t_{dh}, t_{dv})^T - (d_{dh}, d_{dv})^T|_1] \tag{12}$$

where $B$ is the mini-batch with $N$ samples, $s$ is a sample in $B$, $(p_{i1,j1}, p_{i2,j2})$ is a pair of patches randomly picked from the last transformer encoder, $d_{dh}$ and $d_{dv}$ indicate the predicated distance horizontally and vertically, respectively, and $t_{dh}$ and $t_{dv}$ are the true distances.

Cross-entropy is the most common classification loss and can be calculated as follows:

$$L_{ce} = -\frac{1}{N} \sum_{s \in B} log \frac{exp(y_{(s,label_s)})}{\sum_m^M exp(y_{s,m})} \tag{13}$$

where $M$ is the total number of categories, $N$ is the total number of samples, $label_s$ is the true category of sample $s$, and $y_{s,m}$ indicates the probability of a category produced by the network.

The training of the SCM consists of 600 epochs with a batch size of 32. The weights in the SCM are updated by the SGD optimizer. The cosine annealing scheduler with an initial value of 0.0025 and a final value of 0 is employed during training.

## 5. Results

### 5.1. Comparison with CNN-Based SAR Ship Classification Methods

The results of the SCM on the three-category classification task from OpenSARShip are shown in Table 6. For comparison, four other SAR ship classification methods are listed together, including the finetuned VGGNet [13], the plain CNN [16], the group squeeze excitation sparsely connected CNN (GSESCNNs) [17], and HOG-ShipCLSNet [18]. Additional computations outside of the neural networks are necessary in HOG-ShipCLSNet.

**Table 6.** Results of the three-category classification task from OpenSARShip.

| Method | Precision | Accuracy | MAdds | Weights |
|---|---|---|---|---|
| Finetuned VGGNet [13] | 58.72% | 69.27% | $13.84 \times 10^9$ | $15.52 \times 10^6$ |
| Plain CNN [16] | 69.44% | 67.41% | $2.17 \times 10^9$ | $47.44 \times 10^6$ |
| GSESCNNs [17] | 69.56% | 74.98% | — | — |
| HOG-ShipCLSNet [18] | 72.42% | 78.15% | $89.46 \times 10^6$ (Not including HOG and PCA) | $65.11 \times 10^6$ |
| SCM (ours) | 77.74% | 82.06% | $81.46 \times 10^6$ | $0.46 \times 10^6$ |

As shown in Table 6, our proposed SCM achieves the best accuracy, 82.06%, and outperforms all listed single-polarization and dual-polarization methods. Moreover, the SCM has the smallest model size and lowest computational complexity. The results of the SCM regarding the number of weights and MAdds are $0.46 \times 10^6$ and $81.46 \times 10^6$, respectively, which means the SCM is very efficient in computation and storage.

Table 7 displays the classification confusion matrix for the SCM on the three-category classification task from OpenSARShip, indicating that the majority of samples can be correctly predicted. However, the recall of each category is imbalanced; tankers and bulk carriers have the highest and lowest values, respectively. Many other CNN-based SAR networks have reported similar results [15,16,18,19]. The main confusion can be found between container ships and bulk carriers. These two categories have many similarities,

and the quality of samples is poor. Furthermore, the number of samples in the different categories in the test set is imbalanced.

**Table 7.** Confusion matrix of SCM on the three-category classification task from OpenSARShip.

|  | Bulk Carrier | Container Ship | Tanker |
|---|---|---|---|
| Bulk Carrier | 75.61% | 20.73% | 3.66% |
| Container Ship | 12.50% | 82.80% | 4.70% |
| Tanker | 2.05% | 5.48% | 92.47% |

### 5.2. Comparison with Efficient Networks from Computer Vision

To further verify the efficiency of the SCM, four famous light computation models from computer vision are used for comparison. The models used for comparison are the 18-layer residual net(ResNet-18) [51], MobileNetv2 [31], ViT_tiny [23], and CCT_722 [24]. ResNet-18 is a popular small-size general network that has been widely applied in many fields. MobileNetv2 is designed for mobile devices and is widely applied in light computation tasks. ViT_tiny is the first model ranked by computational complexity in the vision transformer family. CCT_772 is a tiny mixture network dedicated to datasets with a small sample size.

As the results of the three-category classification task from OpenSARShip depicted in Table 8 show, the SCM outperforms MobileNetv2 by over 15%. Furthermore, the computational complexity of the SCM is slightly higher than MobileNetv2. Moreover, the SCM requires far fewer weights than the other listed models. In detail, the storage space of the proposed SCM is less than a quarter of MobileNetv2, which is the second-smallest model in the list.

**Table 8.** Comparison with light computation networks on the three-category classification task from OpenSARShip.

| Method | Precision | Accuracy | MAdds | Weights |
|---|---|---|---|---|
| ResNet-18 [51] | 69.40% | 74.64% | $333.10 \times 10^6$ | $11.18 \times 10^6$ |
| MobileNetv2 [31] | 61.80% | 65.83% | $56.65 \times 10^6$ | $2.23 \times 10^6$ |
| ViT_tiny [23] | 46.10% | 64.35% | $208.03 \times 10^6$ | $5.49 \times 10^6$ |
| CCT_722 [24] | 72.29% | 75.66% | $274.07 \times 10^6$ | $4.51 \times 10^6$ |
| SCM (ours) | 77.74% | 82.06% | $81.46 \times 10^6$ | $0.46 \times 10^6$ |

### 5.3. Results on Dual-Polarization

The SCM can also operate dual-polarization classification with high accuracy. The results of the SCM with decision fusion on the three category dual-polarization classification task from OpenSARShip dataset are shown in Table 9. Three CNN-based dual-polarization SAR ship classification methods are listed together for comparison, including the VGGNet with hybrid channel feature loss [14], the Mini Hourglass Region Extraction and Dual-Channel Efficient Fusion Network [15], and SE-LPN-DPFF [19]. SE-LPN-DPFF requires additional computation outside of the neural networks. Dual-polarization classification takes both VV and VH data together as inputs to predict a target. The decision fusion used in the SCM is very simple without additional weights. The SCM outputs two results based on VV and VH, respectively. If the results are different, the result with a higher confidence level will be the final output of the SCM.

**Table 9.** Results of the three-category dual-polarization classification task from OpenSARShip.

| Method | Precision | Accuracy | MAdds | Weights |
|---|---|---|---|---|
| VGGNet With Hybrid Channel Feature Loss [14] | 74.05% | 77.41% | $27.68 \times 10^9$ | $15.01 \times 10^6$ |
| Mini Hourglass Region Extraction and Dual-Channel Efficient Fusion Network [15] | 71.50% | 75.44% | $\geq369.93 \times 10^6$ (Dynamic) | $7.45 \times 10^6$ |
| SE-LPN-DPFF [19] | 76.45% | 79.25% | — | — |
| SCM with decision fusion (ours) | 80.36% | 83.78% | $162.92 \times 10^6$ | $0.46 \times 10^6$ |

### 5.4. Results of Generalization Capability

The computer vision networks compared in Section 5.2 are designed with ImageNet or CIFAR datasets and are well known for their strong generalization capabilities. To verify the generalization capability of the proposed method, the SCM is trained on the four-category classification task from FUSARShip with the architecture seached in the three-category classification from OpenSARShip. Considering the application requirement for computational complexity, the results of the compared computer vision networks are obtained using the same data processing of the SCM and are shown in Table 10. Although the task difficulty has increased, the SCM still has the best accuracy in the list.

**Table 10.** Comparison of light computation networks on the four-category classification task from FUSARShip.

| Method | Precision | Accuracy | MAdds | Weights |
|---|---|---|---|---|
| ResNet-18 [51] | 64.12% | 60.16% | $333.10 \times 10^6$ | $11.18 \times 10^6$ |
| MobileNetv2 [31] | 59.35% | 60.43% | $56.65 \times 10^6$ | $2.23 \times 10^6$ |
| ViT_tiny [23] | 60.76% | 58.29% | $208.03 \times 10^6$ | $5.49 \times 10^6$ |
| CCT_722 [24] | 57.31% | 57.75% | $274.07 \times 10^6$ | $4.51 \times 10^6$ |
| SCM(ours) | 61.95% | 63.90% | $81.46 \times 10^6$ | $0.46 \times 10^6$ |

Table 11 displays the classification confusion matrix for the SCM on the four-category classification task from FUSARShip. SCM can effectively classify bulk carriers and fishing and cargo ships. The tanker category is identified with a low recall and is easily misclassified as another category. We analyzed the confusion matrices of the tested computer vision networks and obtained similar results. The following reason may explain the poor performance in this category. Data processing may remove much information on the tanker category. This work focuses on network architecture; hence, we do not apply additional training methods for small training sets or imbalanced test sets. Many onshore images of tankers can be found in the dataset, which may further decrease the learning. Similarly, no background processing is employed in our method. Lastly, the tanker category in FUSARShip is built with over ten subcategories, including asphalt bitumen, gas, chemical, and crude oil tankers, among others. Some subcategories have strong differences.

**Table 11.** Confusion matrix of the SCM on the four-category classification task from FUSARShip.

| | Bulk Carrier | Fishing | Cargo | Tanker |
|---|---|---|---|---|
| Bulk Carrier | 68.00% | 7.00% | 8.00% | 17.00% |
| Fishing | 3.00% | 70.00% | 13.00% | 14.00% |
| Cargo | 2.00% | 10.00% | 76.00% | 12.00% |
| Tanker | 21.62% | 21.62% | 22.97% | 33.78% |

### 5.5. Ablation Study

A series of ablation experiments were conducted with the three-category classification task from OpenSARShip to confirm the positive effect of each proposed improvement in the SCM. For fairness, all networks used were trained with the same script covering data augmentation, the optimizer, and the learning rate. Table 12 displays the overall results—the conclusion can be drawn that each improvement results

in an accuracy increase. In the cell type column, N and R indicate a searched normal cell and a reduction cell, respectively, and C represents the proposed ConvFormer cell. The details of each improvement can be seen in Sections 5.5.1–5.5.4.

**Table 12.** Results of each improvement in the SCM.

| Method | Cell Type | Accuracy |
|---|---|---|
| PC-DARTS [20] (Baseline) | NNRNRN | 75.66% |
| +PDA | NNRNRN | 80.11% |
| +PDA +MBCONV | NNRNRN | 80.50% |
| +PDA +MBCONV +Transformer Classifier | NNRNRN | 81.59% |
| +PDA +MBCONV +Transformer Classifier +ConvFormer Cell | NRRNCC | 82.06% |

### 5.5.1. Progressing Network Architecture Searching

Table 13 shows the ablation study on the proposed PDA-PC-DARTS. In this experiment, other improvements, the efficient blocks, the transformer classifier, and the ConvFormer cell are used in a target network searched by the original PC-DARTS. As can be seen, employing PDA-PC-DARTS for searching results in higher accuracy. This is because the original PC-DARTS fills its target network with a lot of weight-free operations, leading to weak feature extraction capability.

**Table 13.** Effectiveness of progressive searching.

| Searching Algorithm | Accuracy |
|---|---|
| PDA-PC-DARTS | 82.06% |
| PC-DARTS [20] | 80.81% |

### 5.5.2. MBCONV Block

Table 14 shows the ablation study on replacing DSCONV blocks with MBCONV blocks. A target network based on DSCONV is searched by PDA-PC-DATRS. Then, the transformer classifier and ConvFormer cells are appended under the rule mentioned in Section 3. MBCONV has a stronger feature extraction capability and produces higher accuracy.

**Table 14.** Comparison of DSCONV and MBCONV.

| Basic Block | Accuracy |
|---|---|
| MBCONV | 82.06% |
| DSCONV | 79.41% |

### 5.5.3. Compact Transformer Classifier

Table 15 shows the results of the ablation study on the transformer classifier. A popular linear classifier with global average pooling is listed for comparison. The spatial awareness and learning capability of the transformer classifier boost the performance.

**Table 15.** Comparison of classifiers.

| Classifier | Accuracy |
|---|---|
| Compact transformer classifier | 82.06% |
| Linear classifier | 78.39% |

5.5.4. ConvFormer

Table 16 shows the results of the ablation study on the proposed ConvFormer Cell. Firstly, we replace the ConvFormer cells with normal cells to build an SCT. As discussed in Section 3.4, the SCM does not follow the rule of DARTS to assign the type of each cell. Hence, another SCT is built with the DARTS rule. From Table 16, changing the assignment of cells has a negative effect. However, the ConvFormer cell has more advantages to make up for this negative effect drop and achieves the best accuracy in the table.

**Table 16.** Comparison of different arrangement of cells.

| Network | Cell Type | Accuracy |
|---------|-----------|----------|
| SCM | NRRNCC | 82.06% |
| SCT_1 | NRRNNN | 81.27% |
| SCT_2 | NNRNRN | 81.59% |

## 6. Discussion

### 6.1. Practical Application Scenario

To meet the increasing requirements for ocean surveillance, increasingly more SAR-equipped small aircraft, such as satellites, balloons and drones, are deployed for large area detection and monitoring. Considering security and their energy consumption, application-specific integrated circuit (ASIC) solutions are more suitable for these small aircraft. However, ASIC platforms have many strict requirements for neural networks, especially for model size. Using external memory leads to a large latency in data input/output. The proposed network has the advantages of small model size with good trade-off between accuracy and computational complexity. ASIC implementation can be more flexible than for a CUDA graphics processing unit (GPU), which means ASIC is recommended for the complex prediction graph of the proposed network.

### 6.2. Trade-Off between Accuracy, Number of Weights, and Computational Complexity

Compared with many state-of-the-art SAR classification methods, the proposed SCM has relatively much smaller computational complexity. The proposed network achieves a good trade-off between accuracy and computational complexity. However, we admit that the computational complexity of SCM is slightly higher than MobileNetv2. The efficiency operations, node style prediction graph, ConvFormer cells, and transformer classifier mean that SCM has better accuracy with many small operators. Each small operator is computed with the total feature maps, which leads to increased complexity. To outperform networks for mobile devices both on computational complexity and performance, the operations and cells in the SCM should be further optimized with advanced methods, such as fusion of operators.

### 6.3. Adaptability of the ConvFormer Cell

As we described, only normal cells can be used to build ConvFormer cells which are appended to the DARTS cells. The ablation study showed that breaking the DARTS rule to assign convolutional cells is not recommended. It would be meaningful to design novel ConvFormer cells which contain both normal-type and reduction-type cells. In addition, replacing all DARTS cells with ConvFormer cells should be considered.

### 6.4. Imbalance of Performance over Different Categories

The proposed method has high average accuracy, but the performance of each category is imbalanced. Furthermore, the performance changes in different radar production and application scenarios. Offshore scenarios have higher accuracy than onshore scenarios. This work focuses on network architecture and does not consider additional methods for small training sets and unbalanced test sets. Meanwhile, preprocessing the background may improve the performance of some categories of ships. To obtain better performance of

the SCM over more categories and application scenarios, some special preprocessing and training methods should be studied in future work.

## 7. Conclusions

In this paper, a searched convolutional Metaformer, SCM, is proposed to classify SAR ship images. Firstly, PDA-PC-DARTS, which is designed for SAR datasets with a small data size, is proposed. Cells with strong feature extraction capability can be searched by PDA-PC-DARTS. Secondly, the basic block of operations used in NAS was changed from DSCONV to MBCONV, which results in better accuracy. At the same time, NAS, a CNN, and the transformer are integrated to further improve the learning capability by employing the transformer classifier. In addition, a ConvFormer cell is proposed to further increase the accuracy. The experimental results show that the SCM has many advantages. On the performance side, the SCM achieves state-of-the-art accuracy. Moreover, both the computational complexity and the number of weights of the SCM are very low.

**Author Contributions:** Conceptualization, H.Z., W.S. and S.G.; methodology, H.Z.; software, H.Z.; validation, H.Z., L.X., S.G. and W.S.; formal analysis, H.Z.; investigation, H.Z.; data curation, H.Z.; writing—original draft preparation, H.Z.; writing—review and editing, H.Z., L.X., S.G. and W.S.; visualization, H.Z.; supervision, S.G.; project administration, W.S.; funding, S.G. All authors have read and agreed to the published version of the manuscript.

**Funding:** This work was supported in part by the National Natural Science Foundation of China under grants 62001227, 62001232, and 61971224.

**Data Availability Statement:** Data are contained within this article.

**Acknowledgments:** The authors thank the reviewers for their help with the article during the review process.

**Conflicts of Interest:** The authors declare no conflict of interest.

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
