# Peer review of "SCM: A Searched Convolutional Metaformer for SAR Ship Classification"

_remotesensing, doi:10.3390/rs15112904_

Round 1

Reviewer 1 Report

This draft paper introduces ship classification using SAR images and limitations of traditional methods. It then proposes an approach called Searched Convolutional Metaformer (SCM). The paper explains the progressive data augmentation (PDA) PC-DARTS algorithm, which improves the performance of the target network. It also justifies the use of EfficientNet version Inverted Residual Block (MBCONV) and the integration of a transformer classifier in the searched convolutional Transformer (SCT) for enhanced accuracy. Experimental results demonstrate that SCM outperforms existing methods in terms of accuracy, model size, and computational complexity. The conclusion suggests optimizing SCM's computational complexity in the future. 

Overall, the article provides a comprehensive introduction, detailed discussion, and conclusive remarks about the proposed SCM for SAR ship classification. The paper effectively presents the methodology, justifies the choices made, and highlights the advantages of SCM over existing methods.

One minor issue: please explain the abbreviation of QKV that appears on line 212. 

One minor issue: please explain the abbreviation of QKV that appears on line 212. 

Reviewer 2 Report

Explain the literature section in a comparative manner. Add merits and demerits in the tabular form for better explanation.

Discuss the real time/ practical application of the paper. What is the significance of this study?

Add Materials and Methods section after Literature review. Discuss about the band of SAR data used. Discuss about other SAR bands for more information and knowledge of readers. Take help from below paper:

https://www.spiedigitallibrary.org/journals/journal-of-electronic-imaging/volume-32/issue-2/021609/Review-on-nontraditional-perspectives-of-synthetic-aperture-radar-image-despeckling/10.1117/1.JEI.32.2.021609.full?SSO=1

Explain in detail what makes your work unique.

Reviewer 3 Report

The paper presents a network, SCM, for SAR ship classification. The method is basically kind of combination or integration of some existing networks or techniques, like MBCONV, PC-DARTS, CCT, Metaformer, and etc. Experiments are conducted but lack of persuasion.

Deficiencies and Problems:

(1)    The overall network in Figure 2 is too coarse, giving too information about the proposed method.

(2)    In Figure 3(b), there are “Efficient Conv” and “Efficient Reduced Conv”. Are they different and what’s the meaning by “reduced”? Basically, I don’t think they are two different things.

(3)    The authors state that their method is proposed for SAR ship classification and that the proposed searching algorithm is designed for SAR data. But they didn’t explain how their method is particularly designed for SAR data.

(4)    From Table 6, one can see that the classification precision of three categories is severely imbalanced, which is not desirable in most applications. Same problem can be found in Table 9, where Tanker only obtained 14.86%. While the authors gave not any analysis, and claimed that “ SCM can effectively classify ……” (Line 460-461).

(5)    English Language need to be greatly improved, and spelling and syntax errors should be carefully checked. Just name a few:

(i) In Line 37, “SAR date” should be “SAR data”.

(ii) In Line 234 and 235, “All the efficient operations use MBCONV …….” is ambiguous.

(1)    English Language need to be greatly improved, and spelling and syntax errors should be carefully checked. Just name a few:

(i) In Line 37, “SAR date” should be “SAR data”.

(ii) In Line 234 and 235, “All the efficient operations use MBCONV …….” is ambiguous.

Reviewer 4 Report

Authors present a searched convolutional metaformer for SAR ship classification in this manuscript and comments are listed as follows. There exists 6 major issues and 2 minor issues.

Major issue 1: Authors should also use Mean Average precision (MAP) to analyze experimental results in this manuscript.

Major issue 2: The concept of resolution of SAR images must be clarified. Description such as ‘resolution of input images is 76×76’is wrong. Authors must clarify different concepts of resolution in optical images and SAR images.

Major issue 3: As described in 4.2, samples from FuSARship are resized from 512×512 to 128×128. In this case, the performance of classification will degrade. Why can the proposed method have better performance?

Major issue 4: As described in the abstract, the main highlight of the proposed method is lower computational complexities. Authors should present a table which contains all modifications in this manuscript which lead to lower computational complexities.

Major issue 5: It is common that lower computational complexities will lead to the loss of performance. Why can the proposed method have better performance and lower computational complexities at the same time?

Major issue 6: In Table 5, inputs of different methods are not the same. Some inputs are VV or VH, and some are VV and VH. In comparison, inputs of different methods must be the same.

Minor issue 1: All the abbreviations should appear with full name which there are firstly presented in this manuscript. For example. ‘NAS’,’PDA-PC-DARTS’,’VGGNet’, ‘MADDs’ and so on.

Minor issue 2:Units of MADDs and Weights should be also described. i.e. description of ‘B’,’M’

English writing needs moderate editing.

Round 2

Reviewer 3 Report

The authors have answered my concerns properly.

Reviewer 4 Report

I think this paper has been revised carefully according to the comments from reviewers. This manuscript can be accepted for publication in this version. 

Minor editing of English language required